# Association of Physical and Emotional Parameters with Performance of Firefighters: A Systematic Review

**DOI:** 10.3390/ijerph21081097

**Published:** 2024-08-19

**Authors:** Vinícius Montaguti Farinha, Edilson Fernando de Borba, Poliana Piovezana dos Santos, Anderson Zampier Ulbrich, Evaldo José Ferreira Ribeiro, Marcus Peikriszwili Tartaruga

**Affiliations:** 1Physical Education Department, Federal University of Paraná-UFPR, Curitiba 81530-000, PR, Brazil; vinimontaguti@hotmail.com (V.M.F.); borba.edi@gmail.com (E.F.d.B.); zampier@ufpr.br (A.Z.U.); evaldo.ribeirojr@ufpr.br (E.J.F.R.J.); 2Human Movement Sciences, University of the State of Santa Catarina-UDESC, Florianópolis 88035-901, SC, Brazil; poliana.pds25@edu.udesc.br; 3Physical Education Department, Midwestern Parana State University-UNICENTRO, Guarapuava 85040-167, PR, Brazil

**Keywords:** firefighter performance, firefighting physical ability test, mental demands, physical demands

## Abstract

Firefighting requires a high level of physical fitness and causes substantial psychological stress, engendering musculoskeletal, mental, and cardiac issues. Consequently, it is necessary to measure the preparation of the firefighters daily through the Firefighting Physical Ability Tests (FPATs). According to the literature, some variables are more important for performance in the FPAT. Therefore, we aimed to summarize evidence that relates physical and mental aspects to the FPAT performance. We used the Preferred Reporting Items for Systematic Reviews and Meta-Analysis (PRISMA) method, screening 1055 records from databases and selecting 15 that met inclusion criteria. No emotional and psychological variables were correlated with the FPAT. Most research shows significant correlations between the FPAT performance and the following: aerobic fitness, upper body endurance and strength, anaerobic capacity, body fat, and age. Lower body endurance and strength, as well as anaerobic power, had a low number of investigations and need to be further explored. Abdominal endurance showed weak correlations, while flexibility did not show any correlations in most studies, although these should be considered for injury prevention. We recommend that fitness programs and evaluations include a global analysis considering the evidence presented for methodological improvements.

## 1. Introduction

Firefighting is an occupation that requires a high level of physical fitness calling for different abilities from typical jobs [1,2,3]. Due to distinct kinds of emergencies attended by these professionals, involving various patterns of movement and unhealthy situations, in addition to wearing personal protective equipment (PPE) and self-contained breathing apparatus (SCBA), thus adding a total weight of approximating 23 Kg, an excessive physical and physiological burden is put on the firefighter while on duty [4]. Firefighting, vehicle extrication, pre-hospital care, hazardous materials, search and rescue, disaster assistance, aquatic rescues, and others are some examples of the activities of these employees [5,6].

Furthermore, firefighters are often exposed to situations that may cause enormous psychological, emotional, and mental stress, including hormonal changes such as increased levels of plasma cortisol [7,8]. This extreme metabolic loading and stress, whether acute or chronic, can lead to various types of musculoskeletal injuries, a decrease in mental health, cardiac events, respiratory abnormalities, or even sudden death [9]. In this scenario, the question of how well prepared the firefighter is, whether physically or mentally, is emerging.

Some studies are categorical in affirming that some physical variables are more important than others for performing daily occupational tasks for the military population [10]. Worldwide, many tests aim to measure firefighter readiness and preparedness for the job [1,11,12,13,14]. Such tests involve the main tasks performed by firefighters while attending an occurrence. Some examples are the Candidate Physical Ability Test, Trondheim Test, Pack Hike Test specific for wildland firefighters, Fire Fit Test, and others [15,16,17,18,19]. These tests are often called Work Performance Examinations (WPE) or Firefighting Physical Ability Tests (FPATs). They usually comprise a few tasks performed in a circuit mode that simulates the conditions faced by these professionals, with most of them rated in total time scored.

A recent review has observed the relationship between fitness and WPE in firefighters, analyzing results in field tests as predictive values of the work performance examinations [20]. Upper body strength and endurance were summarized as the strongest correlations to the FPAT. In other studies, aerobic fitness was evaluated with a 1.5-mile run and 12-min test and had weak to moderate correlations with the WPE (r = 0.38 and r = −0.62, respectively) [21,22]. Moreover, the cardiorespiratory component was assessed by quantifying maximal oxygen consumption (VO_2max_) through laboratory tests and showed strong correlations with job-related tasks [23]. Additionally, variables such as age, body composition, anaerobic power, and others have also been correlated with performance, showing different values [16,24,25,26].

Therefore, regarding these correlations, we highlight that some physical variables have been studied more than others. For anaerobic power, for example, there is limited evidence, and when correlated with FPATs’ performance showed varying results (r = 0.79 [22] and r = 0.4 [19]). In contrast, aerobic fitness was assessed more frequently [21,24,25,26] and showed significant correlations with WPEs, although the “r” values differed (r = 0.38 [21]; r = 0.32, not significant [22]; r = 0.7 [23]; r = 0.36 [24]; r = 0.53 [26]).

Apart from that, recent literature has shown that mental, emotional, and psychological variables are related to performance in sports [27,28,29]. A self-talk motivation, for example, can elicit a better performance in 10 Km time-trial cycling when compared with neutral self-talking [30]. High levels of stress, leading to mental fatigue or the diagnosis of burnout, were also associated with poor performance or injuries in various sports [28,29,30,31]. In light of this, public safety occupations, including firefighters, have been considered as tactical athletes or tactical populations [32,33].

This fact might be explained by the execution of highly physically demanding tasks combined with elevated levels of mental and psychological load [32], considering they must be ready to face any manner of threat, whether physical, environmental, or psychological. These links between physical and mental aspects have been recognized as interrelated factors of health and performance for athletes, evaluated as similar levels of need in both contexts (sports performance and firefighting) [33]. The above-mentioned supports the notion that firefighters and other occupations, such as military personnel, police, or paramedics, should be evaluated and trained as tactical athletes [33].

Therefore, taking into account the relation between mental and physical demands, it is reasonable to understand that these factors could also have an impact on the performance of firefighters, since they are included in the tactical population.

This review aims to understand the relationship between physical, mental, psychological, and emotional variables and the performance in WPE among firefighters. Previous research has included field tests, considering their more accessible application [20]. Furthermore, there is a lack of systematic reviews on this topic already stated elsewhere [34]. Hence, exploring the above-mentioned relationship is necessary and would help to elucidate the theme, since it has not been thoroughly covered. We understand that a wide range of variables, including laboratory tests, psychological variables, and others, can promote a deeper knowledge and holistic approach of the performance of firefighters in job simulations. As a final point, we highlight that the study protocol was registered in the International Prospective Register of Systematic Reviews (PROsPERO CRD42022375167).

## 2. Materials and Methods

### 2.1. Search Procedures

This systematic literature review was developed by the Preferred Reporting Items for Systematic Reviews and Meta-Analysis (PRISMA) guidelines [35,36], and the primary search process lasted from October 2022 to August 2023. The selection was conducted in two phases and independently by two researchers (VMF and EB). The first phase corresponds to reading the title and abstract of all references identified in the databases. Studies that meet the eligibility criteria and those in doubt were read in full in the second phase.

Disagreements between reviewers were resolved by consensus and, when necessary, through a third reviewer (MPT). In this scene, fourteen of the fifteen included studies were in consensus between the two researchers, achieving an agreement of 93%. A total of 8 papers, of the 142 screened, were selected as in doubt by the two reviewers (6 by VMF and 2 by EB). From these eight, after consulting the third reviewer, one of them was included because it met the inclusion criteria. Article language was restricted to English, Portuguese, and Spanish. Results were exported to a single file in the EndNote software.

Databases used in the search were PubMed, Scopus, SportDiscus, and Embase with the following terms applied: ((firefighter OR firefighting OR firefighter OR fire rescue OR public safety) AND (test OR evaluation OR task OR exercise OR performance OR activity OR emergency OR ability OR simulation OR demands OR physical OR cognitive OR stress OR burnout OR motivation OR wellbeing OR fatigue OR mental OR emotional OR psychologic OR chronic OR recuperation OR rating of perceived exertion OR humor OR respiratory tract infection). Search criteria are available in the Appendix A. Figure 1 summarizes the primary process of searching.

### 2.2. Eligibility Criteria

The eligibility criteria followed the population, intervention, comparison, and outcome (PICO) method [35,37]. Studies that considered firefighters on active duty, whether volunteers or careers, were included in the population. Research that used recruits with no experience and civilians were excluded due to the bias of experience in job tasks. For the intervention process, articles that applied tests that measured performance in job-related firefighting tasks were included, with the parameters being time or other physiological factors. Excluded researches were evaluations in which firefighters did not use complete PPE; used simulated weight vests, considering the bias of specific properties of the gear; applied tests only for one particular group of professionals such as wildland or pre-hospital care; used only one task, like only stair climb or only forcible entry, for example; applied live burn scenarios, since hot environments can influence. Considering comparison in PICO method, included were physical field or laboratory tests, physiological and anthropometric parameters, and emotional, mental/cognitive, or psychological variables in any measure. Regarding outcomes, the search included reports that showed values of statistical correlations, in any method, between the job performance tests and the comparison parameters.

### 2.3. Data Extraction

After the selection of eligible full-text research, data were manually extracted by one reviewer (VMF) and checked by another reviewer (EB). A Microsoft Excel spreadsheet was used to summarize information, including qualitative variables and averages plus standard deviations for the quantitative ones. Initially, the following variables were gathered: author, goals, year, and the environment where data were collected. Next, we collected population characteristics, the total sample size, sex, age, height, body mass, percent fat mass, and years of service of these professionals. For intervention, we described the WPE tests, tasks applied, adopted protective gear, manner of execution (individual tasks or circuit mode), total time and time of each individual task, including physiological variables measured during tests. For the comparison, all tests used in studies were reported (psychological, mental, cognitive, physical, and physiological). In addition, the fitness test results were congregated as specified measurements (time, distance, score) or physiological effects such as VO_2max_, anaerobic threshold, and cardiac frequency. In addition, values of physiological variables per se were included. All the statistics applied in the research were detailed. Finally, for outcomes, meeting the main objective of this review, we show correlations between the predictive variables and performance in WPE.

### 2.4. Quality Assessment

For quality appraisal of studies, we observed the Physiotherapy Evidence Database Scale (PEDro). Given this scale has been widely used in previous reviews [20,37,38] as a validated tool, and was specifically designed for physiotherapy and physical performance studies, we judged it as particularly well suited for evaluating research on the theme of this study, allowing a consistent quality assessment [39]. The examination was also conducted separately by two reviewers (VMF) and (EB). After rating the studies, a comparison was made between researchers and then any discrepancy was discussed to reach an agreement. The initial ratings scored for each reviewer showed a consensus of 80%, with 12 papers being equally classified.

This scale has 11 questions with “yes” or “no” answers that evaluate the validity of the research [39]. Items five, six, and seven were excluded from the evaluation since the design of the studies did not incorporate these methods. Hence, the maximum achievable score was seven, considering that item one is excluded from the calculation. Studies were classified as 6–7 (excellent), 5 (good), 4 (moderate), and 0–3 (poor). Previous literature in exercise science has used this method to adjust reports’ ratings [20,38,40].

## 3. Results

### 3.1. Search Results

The search criteria have taken us to a result of 15 studies [19,20,21,26,41,42,43,44,45,46,47,48]. Figure 1 shows the details of the screening, which followed the Prisma Flow Diagram. All the reports had a score of a minimum of 4 (moderate) on the modified PEDro scale, three of them with five items counted (good) [21,41,47]. All the results of the PEDRo scale are in Appendix A. Table 1 summarizes the main characteristics of the papers. The method to classify physical tests was used before [20] and judged as good for a better understanding of results.

Interestingly, no article that related emotional, mental, or psychological parameters with a performance firefighter test or WPE was found according to the established criteria. Thus, only physical, physiological, and anthropometric variables were included.

### 3.2. Fitness Tests

The tests utilized were laboratory and field tests with indirect and direct measurements. The absolute results of each one are presented in Appendix A. To measure aerobic fitness, nine studies used predictive variables of velocity, heart rate, age, and body mass for estimated VO_2max_. [21,22,24,26,41,42,43,45,48], while five used metabolic measurements and a gas analyzer [23,25,26,44,47]. For muscular endurance, five studies used tests such as pull-ups or chin-ups for upper body endurance [19,21,26,47,48], and three used weight exercises for upper body endurance [22,24,41] and lower body endurance [22,41]. Upper body strength and lower body strength were all measured with weighted exercises or with hand grip strength, except for one that used a specific test for lower body strength [42]. Five studies evaluated anaerobic capacity by applying run tests and step tests [19,22,41,45,47]. Vertical jump was assessed for anaerobic power in two papers [19,43]. Sit-ups were used for abdominal endurance, and only one study measured strength [19]. The sit and reach test was the only method to assess flexibility.

### 3.3. Performance Tests

Although all performance tests had similar interventions considering firefighting activities, they were slightly different in the number of tasks performed, kind, and whether circuit mode or individual movements. Considering a main function of the activity, like rescuing a victim, we note that all the studies included this movement pattern, in any mode. At least nine articles applied the three most common actions: stair climbing, victim rescue, and hose handling [19,21,22,41,43,44,45,46,47]. Four protocols were conducted individually or with rest among movements [22,42,47,48], and all others were sequentially performed. Circuit tests with specific protocols resulted in different performances, ranging from 96.7 ± 23.0 to 610 ± 79 s [23,43]. It is explained mainly by the number of tasks, kind, and distance spent in the execution of the distinct movements included in each protocol.

### 3.4. Correlations between Fitness Tests and Performance Firefighting Tests

Appendix A shows the details of outcomes and a summary of the variables that had a statistical significance considering the results of fitness and performance tests. Most of the studies used the Pearson correlation coefficient to measure the degree of relationship, but two used other methods [26,48]. The results of fitness tests were primarily related to the performance time of the total circuit or the time to complete individual tasks. Two studies included other performance measures besides time, such as lactate concentrations and heart rate [26,45].

Considering aerobic fitness, two articles did not provide this measure. Of the 13 others, 2 did not find significant correlations between aerobic fitness and results in the performance test [22,25]; all the 11 others had a significant correlation. For upper body endurance, eight had any measure [19,21,22,24,26,41,47,48], and one was not correlated with performance time [47], although associated with the individual task time hose advance. Upper body strength was measured in ten of the records [19,21,22,24,26,41,42,43,47,48], being correlated with performance measures in nine of them, except by one [47], only associated with the individual task hose advance.

Two papers applied tests assessing lower body endurance [22,41], one of them correlated with performance time in the FPAT for squat endurance [22]. For strength in the lower body, eight studies included the measure [19,22,24,41,42,43,47,48], and four showed a significant correlation with WPE [41,42,43,48]. Anaerobic capacity was evaluated in five articles and had significant correlations in all the studies [19,22,41,45,47]. One of them was not correlated with time in performance tests but with lactate concentration [45]. For anaerobic power, two researchers applied the vertical jump method, and both elicited significant correlations with the ability tests [19,43].

Abdominal endurance was assessed in seven protocols [19,21,22,26,41,47,48], having statistically significant correlations with firefighting tests in four studies [19,21,26,48]. One paper examined abdominal strength and correlated with performance time in the circuit applied [19]. Flexibility was investigated in five experiments, and only one of them had significant correlations with stair climb tasks [21].

Body fat and other variables were also measured and correlated with performance firefighting tests. Only three articles did not evaluate body fat [25,42,43], although three of the twelve that assessed did not correlate this variable with performance [44,45,46]. Seven studies [19,21,23,24,26,41,47] showed significant correlations between body fat percentage and total time performance. Nine records also correlated age with WPE [19,21,23,24,41,43,46,47,48], and five presented significant correlations with total time performance in firefighter ability tests [19,23,24,46,48], with two being correlated with the individual task’s time [21,47].

## 4. Discussion

### 4.1. Mental and Emotional Aspects

In this section, we corroborate the findings about some variables that did not show any results. Surprisingly, even with the terms used in the search, no matches were found, and no reported relations were observed between WPE and the following: “cognitive OR stress OR burnout OR motivation OR wellbeing OR fatigue OR mental OR emotional OR psychologic OR chronic OR recuperation OR rating of perceived exertion OR humor OR respiratory tract infection”.

Our search and screening identified some possible factors contributing to this finding. The eligibility criteria might have been affected due to the restriction of using only circuit protocols for the FPAT instead of live-fire scenarios and hot environments. Many papers used this kind of performance test with heat to investigate the relationships between firefighting activities and emotional and mental aspects [2,7,49,50,51,52]. In addition, unlike this review in which performance in the FPAT was the main objective, most articles emphasize the emotional, cognitive, and mental aspects, using real firefighting scenarios to understand its influence in these variables, and not the opposite [2,7,49,50,51,52,53,54,55,56,57,58,59].

Many studies demonstrated the importance of mental and occupational health in the profession, highlighting the impact of stress, quality of life, fatigue, and job stressors [55,60,61,62,63,64,65]. Stressful situations or emotional disorders in this population arise due to factors faced in the occurrences, such as scenes with gruesome sights, sounds, smells, tastes, and touches [64,66]. Post-traumatic stress, for example, is a condition that affects 7% of rescuer workers [67], and this prevalence is far lower in the average population [68], underlining the importance of mental health among firefighters.

In addition to mental health issues, although decreases in firefighter performance on the FPAT have not been reported in the literature in a direct correlation manner, we can observe the fact that emotional and mental aspects are being heavily investigated in the sports context and showed influence on performance (impairments or enhancements) [28,30,31,69,70,71]. A self-talk motivation, for example, can elicit a better performance in 10 Km time-trial cycling when compared with neutral self-talking in total time (*p* = 0.033) and oxygen consumption (*p* = 0.018) [30]. In addition, burnout syndrome was seen as a potential reducer in performance among youth athletes, and an early recognition and treatment of depression can lead to improved performance outcomes.

Hence, this comparison is pertinent because this kind of population (emergency responders) has been compared and denominated as “tactical athletes” because of the constant physical and mental training [33,72]. Furthermore, the competitions have been compared to the real occurrences they face on duty [33,72].

Motivational positive and instructional self-talk were seen through a systematic review as positive strategies for performance enhancement in different sports modalities, with cognitive and behavioral aspects being the most related factors [27]. Also, goal setting is capable of yielding improvements in performance in the 2.3 Km running time in high-school runners, with ego orientation (r = 0.39) and task orientation (r = 0.38) both correlated with improvements [28]. In the same way, athlete burnout syndrome was shown as a motive for affective problems such as low mood and hostility or cognitive issues such as distracted focus, memory, and helplessness [31]. Furthermore, physical aspects, such as fatigue, increased probability of injury, absenteeism, and poor sports performance, are problems associated with this syndrome [31].

In addition, mental fatigue has been studied in many sports. It is seen as an impairment in performance due to a mechanism of increasing perception of effort, thereby reducing motivation and requiring the brain to make a more significant effort for the same level of enforcement [69]. A reduction in the 3 Km time trial was showed following a prolonged and demanding cognitive task thereby increasing perception of effort (Glass’s value ∆ = 0.27) [28].

Considering those mentioned above, it is reasonable to understand that mental, psychological, and emotional aspects should be better explored in the firefighter population, especially regarding direct performance measurements.

### 4.2. Physical Aspects

Considering an overview of physical fitness and the FPAT, the most correlated variables were aerobic fitness, upper body endurance, upper body strength, anaerobic capacity, body fat percentage, and age. All the others, except flexibility, did not show strong support for interpretations, although methods deserve a discussion to examine the reasons for these observations.

Aerobic fitness is a physical component mainly used in firefighting tasks [23,73,74], and its level is set as a VO_2max_ minimum of around 42 mL·min^−1^·Kg^−1^ [75,76,77]. In this research, although 11 of 13 have shown significant correlations, all were weak to moderate and were around r = |0.4|. This finding agrees with literature since moderate oxygen consumption is needed. In this review, though, tasks that did not use actual fire scenes were selected, and oxygen consumption could be affected by this hot condition [20,78].

The research of Rhea et al. [22] and Perroni et al. [25] were the only experiments that did not find correlations between aerobic fitness and WPE. A main reason and some secondary reasons might be cited for this happening. Firstly, the method proposed by Rhea et al. [22] applied for the FPAT was not in circuit mode but in individual tasks with 10 min of rest in between, and their objective was precisely this. The aim was to exclude the fatigue factor and analyze the correlation with the task per se. In addition, since each task had a short time to complete and the sum of the total time was 161.8 ± 40.8 s, it is reasonable to infer that aerobic fitness would not have a primary role in the execution of movements.

On the other hand, Perroni et al. [25] had a total time of 704 ± 135 s to complete the WPE in a circuit mode. In this case, the purpose for no relation with aerobic fitness may be because of one specific task described by the author as “Find an Exit”. In the simulation, the firefighter had to complete a maze in a dark room. This task had significant variability (5 to 11 min) and required the firefighter to have a high skill level in orienting due to dark ambient and unknown obstacles encountered. Also, this part of the circuit had a central role in the total time spent. Considering that it was executed at a walking pace and had low aerobic demand [25], other components such as stress, anxiety, and emotional handling might be more critical in this case.

Considering the results of this review, upper body endurance was another variable that appears to have a reasonable degree of importance in firefighter activities. All eight studies analyzed had some degree of correlation, and a considerable number had moderate relationships [22,26,41,48], while others had lower degrees of correlation. Three studies used weighted exercises to measure upper body endurance [22,24,41], and the other used bodyweight movements such as pull-ups and push-ups.

Considering the nature of the firefighter job and the reality of carrying heavy equipment for long periods, weighted exercise might be a better method to assess upper body endurance than bodyweight exercises. However, the last are more accessible for application. Interestingly, two of the strongest relationships encountered were from studies that assessed upper body endurance in the weighted exercises method [22,41]. Using endurance to sustain a required muscular force for several repetitions in one movement is a reality in WPE for firefighters [19,34] and can explain the significant correlations encountered.

Strength in the upper body was also a primary predictor of performance. It might be explained by a reduced effort to perform a given task and better execution in each movement of an FPAT [34]. In addition, in real job tasks, firefighters have to carry and handle heavy equipment such as jaw of life (33 Kg), charged hose (51 to 69 Kg), ladders (25 to 61 Kg), rescue tools for car accidents (36 Kg), and others [43,79]. Hand grip strength was the most assessed test for upper body strength and presented significant correlations in most studies, being the variable with a higher r-value of around 0.6. Although the test shows a general body force compared to specific movements such as bench press, it is easier and can be a field method for strength assessment [21].

While upper body strength has strong evidence of correlations, for lower body strength the results of this study did not support a robust relation between this variable and performance in WPE. One of the records did not report the results of the correlation assessed [24]. Thus, a total of seven studies could be considered. Four found generally weak to moderate significant correlations with performance, and one found a correlation with the hose pull task, r = |0.48| [22]. The strongest of them was found when the result was divided by the total mass of the individual, in the maximum load of squat (r = |0.7|) [41]. On the other hand, another paper used the squat exercise and did not find significant correlations with performance, with r = |0.22| [19]. The leg press exercise and NIOSH lower limb test had significant and moderate to low correlations with performance, with r = |0.395| (leg press) [41], and r = |0.25| (NIOSH) [42].

In two cases, individual tasks such as hose pull and rescue victim were significant [22,41,43]. Only one study found significant correlations with stair climb tasks, r = |0.7| [43], even adopting similar methods and population [22]. The relative maximum load in squat used in this research [43], when divided by total mass, probably had influence on this result, since stair climbing is an activity in which one needs to move his own body toward the stairs.

The relatively low number of records observed might be a factor that impedes accurate outcomes. Previous investigations with other inclusion criteria also differed in conclusions [20,34]. One confounding factor could be explained by the test applied since the One-Repetition Maximum test (1-RM) depends on the experience of weight training, body composition, and others [80]. When the 1-RM squat was assessed, results were inconsistent, and factors like technique and range of motion can have a significant impact [81,82]. On the other hand, when whole body movements were applied (deadlift and back and leg dynamometer) [41,48], significant correlations were shown.

Only two studies showed results for endurance in the lower body and differed in outcomes [22,41]. Rhea et al. [22] found significant correlations in squat endurance (maximum repetitions with 61.4 Kg) with the FPAT (r = −0.47) and hose pull individual tasks (r = −0.56). In contrast, Schmidt et al. [41] found all nonsignificant and poor correlations with the Revised Grinder Test or any individual tasks using leg press endurance (50% of 1-RM). In previous research, this variable was poorly mentioned [20,34]. Differences in the test might explain the results faced in this review since a fixed weight set for squats used in Rhea et al. [22] could vary a lot in terms of the percentage of 1-RM. In addition, squats can be performed using different techniques and should be used cautiously to compare studies [81,82]. Hence, this variable should be better explored considering its relation to performance in the FPAT.

Anaerobic capacity was also a variable with solid evidence about FPATs. From five included studies, four showed moderate correlations with total time in WPEs (“r” ranged from |0.4| to |0.79|), and one found positive correlations between lactate concentration in the 300 m run test and lactate concentration of WPE [45]. This high correlation aligns with other reviews [20], although using fewer studies for comparison. In addition, another review found that sprint tests (100 to 400 m) correlate highly with casualty drag tasks [10]. Other authors did not incorporate this theme [34]. The finding is reasonable since high levels of strength must be sustained for a considerable time (around 30 to 120 s) [21,22,25,42] in different tasks such as victim rescue, hose pull, or equipment carry. Considering that the glycolytic system is in demand in these tests, the anaerobic sprint tests can be an excellent method to assess performance in the FPAT [83]. This variable should be better explored in future research.

In this review, power was collected in only two studies that used the vertical jump test [19,43]. Both found significant correlations. Michaelides et al. [19] found an r = −0.41 with total time in WPE, and Schmit et al. [43] found an r = −0.66 with total time in WPE and also with time in individual tasks: stair climb (−0.67), hose advance (−0.49), and rescue victim (−0.60). As we can see, moderate correlations were shown. Differences in the protocols could explain the lower correlations in the first study since power was related to body mass. These findings can point us to the importance of the phosphagen system because of rapid tasks lasting less than or around 10 s [83]. Hence, little evidence was found, in agreement with the literature, and further research is needed to understand the importance of power and its role in performance in the WPE tests [20].

Endurance in abdominal strength was found to have significant and weak to moderate correlations with performance in WPE in four [19,21,26,48] of six studies assessed [19,21,26,41,47,48]. One article also measured strength in abdominal muscles through an isometric device and found moderate and significant correlations [19]. It seems that abdominal muscles can have moderate importance for predicting the performance in WPE. In addition, it is essential to note that this measure needs future research when considering similar protocols and different results of correlations between studies.

Although the fundamental role of this variable needs to be clarified, its benefits in preventing injuries, pain relief, and improving balance are notable in the scientific literature [38,84,85,86]. This should be considered when planning a training program. Another variable that is a primary factor contributing to muscle health and injury prevention is flexibility [38,72,83,84,86,87]. In this review, flexibility was not related to performance in the FPAT and is in line with previous literature [20,34]. Apart from this, it should be included in a training program because of its benefits.

Body composition, in the measure of body fat percentage, had a majority of evidence supporting a moderate and significant correlation between this variable and performance in the FPAT. Given that body fat percentage (BF%) is a measure of subcutaneous adipose tissue, leaner professionals tend to carry less total weight and less nonfunctional tissue [88]. Hence, there is a tendency for less physical effort and better performance in firefighting tasks. Furthermore, preventing cardiac diseases is a peremptory issue when considering body fat, given that a minor decrease in body fat can reduce the risks of this outcome [83]. Thus, assuming cardiac events are a health issue in this population [89,90,91], it is wise to have a fitness and health program that also promotes the reduction of body fat percentage.

Age was another variable assessed in this review, and significant correlations were found with at least one task being rescue victim and stair climb [21,47] or total time performance [19,23,24,46,48] in seven of nine studies. It is very relevant given that maintaining physical fitness across the entire firefighting career can be very challenging. The loss of positive adaptations acquired in the academy may occur only 24 weeks after leaving the training periods [92]. Additionally, less physical aptitude and performance in the FPAT have been documented in firefighters with aging [26,92]. This should be considered to reorganize personnel in different functions or to plan WPEs and fitness programs.

## 5. Conclusions

This review has several limitations that we ought to list here. We could not ignore the fact that different protocols were used for the FPATs and there was a lack of using live burn scenarios. These kinds of tests were elaborated as an attempt to measure the performance of a firefighter in a context similar to what they face in the real world. A real occurrence has many more variables and all kinds of unpredictabilities. In addition, the technical factor is not fully evaluated in the WPEs as well, considering that most of the exercises in circuit are physically exhausting and do not require a great amount of ability with the tools used. We tried to standardize the tests selected to be the more equally executed, so there is a method for a relative comparison between them. The usage of live burn scenarios would make this difficult.

Considering the psychological variables, the methods used in this review were not sufficient to find any correlations between those and performance in the FPATs. This fact might be seen as a limitation since a wider search procedure would probably show some results. On the other hand, it is notable that there is a need for evaluating the influence that mental and emotional aspects have on the performance of firefighters. In this review, we could show that there is a gap regarding this relationship when using a direct manner of evaluation.

Also, as a strength point, this review could show that it is essential to consider general physical fitness when measuring performance in the FPAT. Some variables have more supporting evidence for being set as a predictor of performance in the FPAT: aerobic fitness, upper body endurance, upper body strength, anaerobic capacity, body fat percentage, and age. Other variables, such as lower body endurance, lower body strength, and anaerobic power, must be more explored because of poor evidence, confounding factors, and different evaluation methods that limit conclusions. Abdominal endurance showed weak to moderate correlations with performance in WPEs, despite its high importance in preventing injuries. Also, flexibility is not correlated with FPAT results. However, just as abdominal endurance, it is a significant variable related to injury prevention.

Regarding performance, we emphasize that due to heavy equipment carried and weight handling, firefighters require upper body endurance and strength and should implement this kind of training in routine. In addition, aerobic fitness is generally practiced among this population, and as seen, it is vital for performance and has a lot of health benefits. On the other hand, sprints are not usually executed and the anaerobic capacity is not assessed in many corporations. Considering its importance in performance in simulated job tasks, this should be included in fitness programs and reviewed for inclusion in physical aptitude tests. Circuit mode training with short durations and specific firefighting tasks has been shown as suitable for this purpose [93]

### Practical Applications

With the results of this study, we can state that aerobic fitness, upper body endurance, upper body strength, and anaerobic capacity should be included in a training program specifically designed to improve firefighters’ performance on the FPATs. Although the papers reviewed were not interventional, there is a substantial body of evidence that supports this finding.

Additionally, tests such as short sprints (400 m) should be included in an evaluation program and in a training routine to improve performance or for assessment purposes. Therefore, we recommend that fitness programs include a comprehensive analysis that considers age, performance on duty, and injury prevention, including stretches and core stability.

## Figures and Tables

**Figure 1 ijerph-21-01097-f001:**
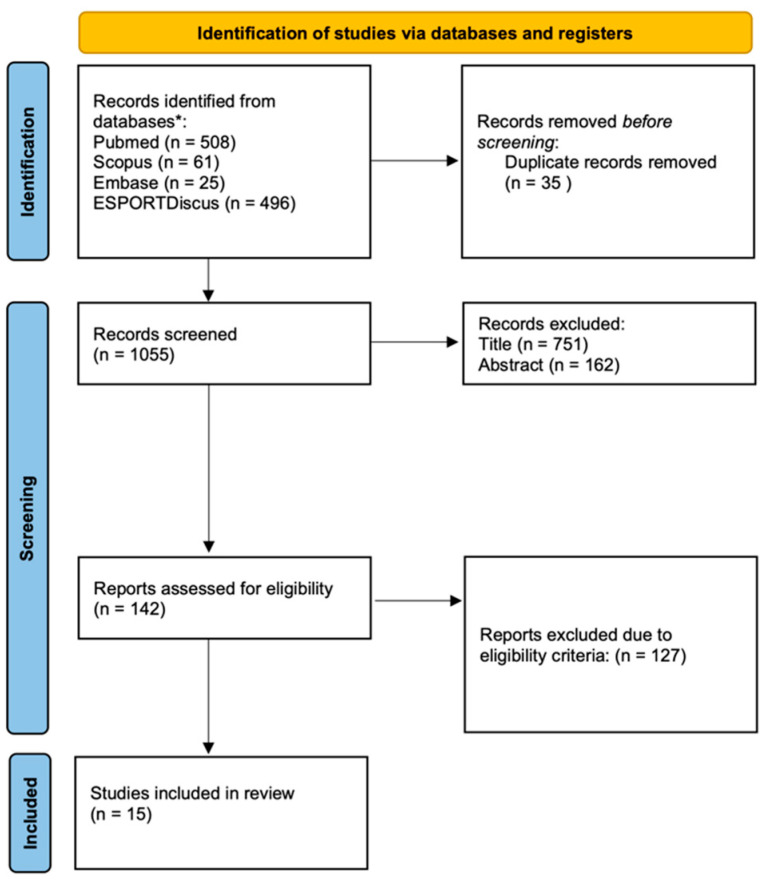
Prisma flow diagram and screening.

**Table 1 ijerph-21-01097-t001:** Descriptive characteristics of studies.

Study	(N)	Characteristics	Fitness Tests	Performance Tests	Description	Turnout Gear	Statistics
Davis et al., 1982 [26]	n = 100 Gender: not informed Age (y): 33.1, range: 21–57 Height (cm): 179.69, range 163.1–190.0 Mass (Kg): 83.44, range 60.2–114.4 Body composition (BF %): 21.1 Years of service: N/A	Experienced professionals	AF = 5 min step test; Balke treadmill test (Beckman Metabolic Measurement) UBe = chin ups (maximum reps) UBs = hand grip strength LBe = N/A LBs = N/A ANc = N/A ANp = N/A ABe = sit ups in 2 min FL = sit and reach BF% = estimated (Zuti and Golding)	1. Ladder extension 2. Standpipe carry 3. Hose pull 4. Simulated rescue 5. Simulated forcible entry	Tasks were performed in the sequence as quickly and as efficiently as possible, each subject breathing from the SCBA and avoiding rests between tasks. The elapsed time to complete each task and the total time to complete all tasks were obtained.	Complete protective equipment ensemble (weighing a total of 24 kg), consisting of helmet, Mine Safety Appliance (MSA) self-contained breathing apparatus (SCBA), boots, and turnout coat.	Six variables of performance (split time for five tasks and average heart rate) were selected through a matrix of correlations. For the six criteria and 20 variables, eigenvalues, canonical correlations, and Chi-square statistics were applied. Two dimensions were found plausible that underlay the six criteria and were interpreted as physical work capacity (factor 1) and resistance to fatigue (factor 2).
Myhre et al., 1997 [24]	n = 222 Gender: 218 male; 4 female Age (y): 30.4 ± 9.3 Height (cm): 178.6 ± 7.6 Mass (Kg): 83.5 ± 13.1 Body composition (BF %): 20.1 ± 6.9 Years of service: N/A	Career fire- fighters representing one Army and seven Air Force Base fire departments	AF = sub-maximal cycle ergometry (estimated from heart rate response) UBe = bench press (80 lb, rate 30/min, maximum reps) UBs = bench press 1RM; upright forearm curl 1RM; upright rowing 1RM; LBe = N/A LBs = leg press 1 RM ANc = N/A ANp = N/A ABe = N/A FL = N/A BF% = hydrostatic weighing	1. Structural search and rescue	The task was performed continuously, and performance time constituted the primary criterion of the simulated rescue task performance. Exercise heart rate (Exersentry or Polar chest electrodes); respiration rate and ventilation were also assessed.	Day uniform: 3.31 Kg. Turnout coat, trousers, helmet, hood, gloves, and boots: 7.58 Kg SCBA (fully charged): 11.34 Kg. Total of 22.23 Kg.	Pearson product-moment correlations were computed to evaluate the relationship between each selected measure and the time required to complete a standardized simulation of a fire fighter rescue task. Least squares multiple regression analysis was used to develop a prediction model for task completion time.
Williford et al., 1999 [21]	n = 91 Gender: male Age (y): 31.69 ± 7.39 Height (cm): 177.29 ± 6.38 Mass (Kg): 83.97 ± 10.86 Body composition (BF %): 13.78 ± 4.31 Years of service: N/A	Professionals	AF = 1.5 mile run UBe = push-ups, pull-ups (maximum reps) UBs = hand grip strength LBe = N/A LBs = N/A ANc = N/A ANp = N/A ABe = sit-ups (maximum reps in 60 s) FL = sit and reach BF% = skin-caliper (3-site)	1. Stair climb 2. Hoisting hose in high 3. Forcible entry 4. Hose advance 5. Victim rescue	Circuit of five job-simulated tasks arranged in the following order: stair climb, hoisting, forcible entry, hose advance, and victim rescue.	Helmet, gloves, boots, turnout coat and pants, and self-contained breathing apparatus (SCBA). The weight of the protective clothing and SCBA was ~ 23 kg.	Pearson product-moment correlation coefficients were calculated between the following: (1) total and individual times and (2) descriptive characteristics and physical fitness measures. Stepwise multiple regression analyses were employed to develop linear equations that would best predict the criterion measures of total obstacle course time (PPA).
Rhea et al., 2004 [22]	n = 20 Gender: 17 male, 3 female Age (y): 34.5 ± 6.1 Height (cm): not informed Mass (Kg): not informed Body composition (BF %): 16.6 ± 3.9 Years of service: 6.1 ± 5.2	Professionals	AF = 12 min run Ube = row endurance (20.5 kg, maximum reps, dominant hand); bench press endurance (45.5 kg, maximum reps); bicep curl endurance (13.6 kg, maximum reps); seated shoulder press endurance (11.4 kg, maximum reps); hand grip endurance (maintain ≥ 25.0 kg, s) UBs = bench press 5RM; hand grip strength LBe = squat endurance (61.4 kg, maximum reps) LBs = back squat 5RM; ANc = 400 m run (s) ANp = N/A ABe = ab curls (maximum reps, no time limit, 30 rep per minute cadence) FL = N/A BF% = bodpod	1. Hose Pull 2. Stair Climb 3. Simulated Victim Drag 4. Equipment Hoist	Four separate job performance tasks were performed, each as quickly as possible and timed for performance score.	Turnout clothing, including boots, pants, coat, and helmet. A standard self-contained breathing apparatus (SCBA) tank (25 kg) was worn during each test; however, the face mask was removed for testing.	Fitness measures were combined into one grand fitness score by adding all test scores together (excluding the 400 m sprint time and percent body fat, which were subtracted) so that higher scores represented greater fitness. The performance measures were also added together to create one grand performance time with lower scores representing superior performance. Pearson product-moment correlation coefficients were calculated between the grand scores as well as each separate test score.
Perroni et al., 2010 [25]	n = 20 Gender: male Age (y): 32 ± 6 Height (cm): 177 ± 6 Mass (Kg): 77.2 ± 8.7 Body composition (BF %): N/A Years of service: N/A	Professionals with minimum 3 years of service	AF = graded incremental treadmill test to exhaustion wearing SCBA (K4b2, Cosmed, Rome, Italy; Accusport Lactate Analyser, Roche, Basel, Switzerland) UBe = N/A UBs = N/A LBe = N/A LBs = N/A ANc = N/A ANp = N/A ABe = N/A FL = N/A BF% = N/A	1. Climb a firemen’s ladder and descend a 3-floor building carrying a 20 kg child dummy (child rescue) 2. Run for 250 m 3. Complete a maze in a dark chamber (find an exit) 4. Run for 250 m	Four tasks were included in the simulated intervention to be completed as quickly as possible and energy cost was evaluated using a portable metabolimeter (K4b2, Cosmed, Rome, Italy).	Complete National Fire Protection Agency standard protective firefighting garments. The total weight of the ensemble was approximately 23 kg.	The individual’s VO_2max_ was correlated to the time required to complete the four firefighting tasks and the whole intervention.
Michaelides et al., 2011 [19]	n = 90 (23 failed to complete) Gender: male Age (y): 33 ± 7 Height (cm): 181.16 ± 6.62 Mass (Kg): 97.04 ± 15.51 Body composition (BF %): 23.05 ± 5.58 Years of service: N/A	Professionals	AF = N/A UBe = push-ups (maximum reps) UBs = bench press 1RM; hand grip strength LBe = N/A LBs = back squat 1RM ANc = step test (60 s) ANp = vertical jump Abs * = isometric device (ABMED) ABe = sit-ups (maximum reps in 60 s) FL = sit and reach BF% = bioelectrical Impedance * unique study that measured abdominal strength	1. Stair Climb 2. Rolled Hose Lift and Move 3. Keiser Sled 4. Hose Pull and Hydrant Hookup 5. Rescue Mannequin Drag 6. Charged Hose Advance	The total time to complete the battery of 6 tests included the time between each task; time started when the firefighters began task 1 and stopped when they finished task 6. Individual task completion times were also recorded. The AT test was administered to all firefighters by the same trained instructors.	Protective gear with a total weight of 22.68 kg.	Pearson product-moment correlation coefficients were calculated among the fitness variable scores and the AT time. In addition, Pearson product-moment correlation coefficients were calculated among the fitness parameters and the individual AT tasks in an attempt to identify their importance on each firefighting task.
Schmidt et al., 2012 [41]	n = 48 Gender: 43 male; 5 female Age (y): 30.15 ± 7.24 Height (cm): 1.71 ± 0.08 Mass (Kg):76.96 ± 12.48 Body composition (BF %): 12.60 ± 6.11 Years of service: N/A	Experienced professional firefighters	AF = Léger UBe = bench press (45 Kg, maximum reps); bent-over row (20 kg, dominant hand, maximum reps); bicep curls (14 kg, maximum reps); seated shoulder press (12 kg, maximum reps); hand grip endurance (25 kg, s) UBs = bench press 1RM; hand grip strength LBe = leg press (50% of 1-RM, maximum reps) LBs = deadlift 1 RM; leg press 1 RM ANc = 400-m run (s) ANp = N/A ABe = abdominal curl (maximum reps in two minutes) FL = N/A BF% = Harpenden caliper	1. Hose pull 2. Stair climb 3. Simulated victim drag 4. Simulated ladder raise 5. Equipment hoist Revised Grinder test: 1. Simulated ladder raise 2. Hose pull 3. Static Jaws-of-Life hold 4. Tyre and sledgehammer test 5. Stair climb 6. Attic Craw 7. Simulated victim drag	Five separate job performance tasks followed by a combination of the tasks (the Revised Grinder Test) were tested. The time (seconds) to complete the Revised Grinder Test was recorded and used as an overall job performance score. A minimum of five minutes rest was allowed between the individual job tasks and a minimum of ten minutes rest before the start of the Revised Grinder Test.	The participants completed all tasks in their specified fire-fighting turnout clothing (9.6 kg), including boots, pants, coat, gloves, and helmet.	Significance of associations between physical fitness measures and the job performance measures were determined using Pearson product-moment correlation coefficients and multiple linear regression (stepwise method). For the multiple linear regression, the Revised Grinder test was set as the dependent variable and the independent variables included the descriptive and physical fitness measures.
Siddall et al., 2018 [23]	n = 68 Gender: 63 male, 5 female Age (y): 41 ± 8 Height (cm): 178 ± 6 Mass (Kg): 85.7 ± 12.9 Body composition (BF %): 19.7 ± 5.6 Years of service: N/A	Operational firefighters	AF = graded uphill running protocol (Cosmed K4 B2) UBe = N/A UBs = N/A LBe = N/A LBs = N/A ANc = N/A ANp = N/A ABe = N/A FL = N/A BF% = bioelectrical impedance	1. Equipment carry 2. Casualty evacuation 3. Hose run	Continuous circuit of three tasks completed on a 25 m shuttle course. Participants were asked to complete test with maximal effort, as quickly as possible while adhering to normal safety regulations. Completion time and rating of perceived exertion were taken at the end of exercise using the Borg scale.	Full personal protective clothing consisting of helmet, shirt, tunic, leggings, boots, and gloves (mass of ensemble: ~8.2 kg). A self-contained breathing apparatus (SCBA; mass: 12.1 kg) was donned for the casualty evacuation section of the simulation and removed prior to the hose run.	Pearson correlation coefficients were used to assess the prediction performance time from VO2ABS andVO2REL. Stepwise multiple regression analysis was conducted to determine which combination(s) of selected variables (age, sex, body mass, height, BF%, FM, LBM/FM) alongside VO2 max best predicted FFST completion time.
Nazari et al., 2018 [42]	n = 49 Gender: 46 male, 3 female Age (y): 33.66 ± 9.19 Height (cm): 1.81 ± 0.08 Mass (Kg): 90.35 ± 13.22 Body composition (BF %): N/A Years of service: N/A	Male and female firefighters: recruits = 5; volunteer = 2; professional = 22; 1st rank firefighter = 10; captain = 10	AF = Modifed Canadian Aerobic Fitness Test’s (mCAFT) UBe = N/A UBs = hand grip strength LBe = N/A LBs = NIOSH lower limb strength ANc = N/A ANp = N/A ABe = N/A FL = N/A BF% = N/A	1. Hose drag 2. Stair Climb with a High-Rise Pack	Firefighters were timed while performing each task using a stopwatch and were asked to carry out both tasks at a work rate typically utilized at a fire scene. Heart rate (bpm), respiratory rates (breaths/min), time (s), rating of perceived exertion (0–10) were collected.	Full personal protective equipment (22.7 kg) and the self-contained breathing apparatus (SCBA) (18.1 kg).	Pearson’s correlation coefficients (r) were calculated between firefighters’ physical fitness parameters and simulated functional task completion times. To predict functional task completion times (our dependent variable) in firefighters, we conducted two separate multivariable enter regression analyses: one for hose drag task and the second for the stair climb with a high-rise pack task.
Schmit et al., 2019 [43]	n = 20 Gender: male Age (y): 37.8 ± 8.4 Height (cm): 182.1 ± 7.0 Mass (Kg): 95.6 ± 8.9 Body composition (BF %): N/A Years of service: N/A	Career firefighters between the ages of 24–55	AF = 3 Min Step Test HR UBe = N/A UBs = bench press 1 RM; grip strength LBe = N/A LBs = back squat 1 RM ANc = N/A ANp = vertical jump ABe = N/A FL = N/A BF% = N/A	1. Stair Climb with high-rise pack 2. Charged Hose Advance 3. Victim Rescue Randy	Individual and total time tasks were measured. Tasks were performed sequentially.	Full firefighting gear consisting of helmet, hood, pants, coat, gloves, boots, and air pack (22.7 kg).	The association between the fitness variables and the performance variables were conducted with Pearson correlation coefficients (r).
Stevenson et al., 2019 [44]	n = 69 Gender: 64 male; 5 female Age (y): 48 ± 8 Height (cm): 178 ± 6 Mass (Kg): 85.8 ± 12.8 Body composition (BF %): 19.7 ± 5.5 Years of service: N/A	Operational firefighters from seven UK Fire and Rescue Services (FRS)	AF = maximal treadmill protocol with portable breath-by-breath gas analyzer UBe = N/A UBs = N/A LBe = N/A LBs = N/A ANc = N/A ANp = N/A ABe = N/A FL = N/A BF% = bioelectrical impedance	1. Equipment carry 2. Casualty evacuation 3. Hose run	Participants were asked to complete the test in the fastest time possible while adhering to standard operating procedures, manual handling, and safety regulations. The time taken to complete each of the three stages/tasks was recorded, as well as perceived exertion at the end.	Full firefighting ensemble (i.e., tunic, leggings, boots, flash hood, helmet, gloves [total mass 8.2 kg]), while carrying, but not breathing on, a self-contained breathing apparatus (SCBA) set (total mass 12.0 kg) during the casualty evacuation component of the simulation.	Pearson correlation coefficient was used to determine the relationship between the performance time and VO2 max. Standard error of estimate (SEE) statistics were calculated to determine the size of the mean error from the estimation plot.
Lessa et al., 2020 [45]	n = 10 Gender: male Age (y): 34,2 ± 5,8 Height (cm): 175,9 ± 6,9 Mass (Kg): 78,7 ± 6,8 Body composition (BF %): 13,2 ± 3,3 Years of service: N/A	Male firefighters within the age range between 28 and 50 years old who have experience in firefighting activities	AF = maximum incremental test (Léger) UBe = N/A UBs = N/A LBe = N/A LBs = N/A ANc = 300 m run (s) ANp = N/A ABe = N/A FL = N/A BF% = Cescorf caliper	1. Tower climbing 2. Hoisting hose 3. 40 m run 4. Forced entry and rescue 5. Use of the hose	Firefighter Combat Challenge composed of five sequential stages (Tsim). The total time to execute the test was timed from the sound signal to start the specific exercise until the firefighter dropped the target. Maximum heart rate at the end of the specific simulated test (HRpeakTsim); the total time of the test and the blood lactate concentration level ([Lac]Tsim) three minutes after the end of the Tsim were collected.	Personal and respiratory protective equipment, weight not informed.	The associations between the aerobic and anaerobic fitness assessment variables versus the specific simulated test variables were tested using the Pearson correlation coefficient.
Saari et al., 2020 [46]	n = 64 (stratified by age) Gender: male Age (y): younger 31.8 ± 3.42; older 44.65 ± 5.18 Height (cm): younger 179.85 ± 6.32; older 182.23 ± 5.57 Mass (Kg): younger 92.61 ± 8.73; older 89.77 ± 23.06 Body composition (BF %): younger 15.94 ± 4.31; older 19.49 ± 4.58 Years of service: N/A	Structural firefighters	AF = N/A UBe = N/A UBs = N/A LBe = N/A LBs = N/A ANc = N/A ANp = N/A ABe = N/A FL = N/A BF% = bioelectric impedance	1. The High-Rise Pack Carry 2. Hose Hoist 3. Forcible Entry 4. Hose Advance 5. Victim Rescue	Scott Firefighter Combat Challenge. The tasks were performed sequentially and without recovery.	Personal protective equipment, including a helmet, pants, overcoat, gloves, and boots. A self-contained breathing apparatus was used.	The median age (37 years) within the sample was used to stratify the firefighters into old vs. young cohorts. If significant differences were identified between age cohorts, Pearson product-moment correlation analysis was used to support the differences identified between groups. obs.: Two variables were correlated with performance time, since they were the only ones that presented significant differences between groups: age and body fat percentage.
Skinner et al., 2020 [47]	n = 42 Gender: male Age (y): 38.4 ± 7.6 Height (cm): 180.2 ± 6.6 Mass (Kg): 81.9 [78.1–99.2] Body composition (BF %): 21.5 ± 4.6 Years of service: N/A Mass data presented as median [interquartile range].	Aviation Rescue Firefighters (ARFF)	AF = Incremental exercise on a motorized treadmill UBe = push-ups (maximum reps) UBs = bench press 3 RM; grip strength LBe = N/A LBs = leg press 3 RM ANc = anaerobic step test ANp = N/A ABe = abdominal curl (maximum reps) FL = sit and reach BF% = dual energy X-ray absorptiometry	1. Hose drag 2. Dummy drag 3. Stihl saw hold 4. Stair climb 5. Simulated ARFF emergency protocol	Each task was completed in accordance with established ARFF procedures and separated by a 5 min rest period. Time was used to measure performance.	Full turnout gear, which included 4.5 kg of clothing and 12 kg of breathing apparatus.	Pearson’s or Spearman’s correlation coefficients were used to examine the correlation between measures of physical fitness and test performance. Linear regression was used to estimate the explained variance in the outcome for each of the explanatory variables. To explore which combination of variables best explained the variance in the performance test, a stepwise approach was used, in which the variable with the highest correlation was added first.
Ras et al., 2023 [48]	n = 268 Gender: 239 male; 29 female Age (y): 36 (p25, 29.0–p75, 46.0) Height (cm): 173.5 (p25, 169.1–p75, 178.3) Mass (Kg): 81.0 (p25, 72.5–p75, 89.9) Body composition (BF %): 19.5 (p25, 14.3–p75, 26.1) Years of service: 12.0 (p25, 4.0–p75, 19.0) p25 = 25° percentile p75 = 75° percentile	Full time firefighters	AF = estimated (age, resting heart rate, body mass) UBe = push-ups (maximum reps in 60 s) UBs = grip strength LBe = N/A LBs = back and leg strength dynamometer ANc = N/A ANp = N/A ABe = sit-ups (maximum reps in 60 s) FL = sit and reach BF% = bioelectric impedance	1. Step-ups 2. Charged hose drag and pull 3. Forcible entry 4. Equipment carry 5. Ladder raise and extension 6. Rescue drag	Firefighters were allowed 20 s of recovery between tasks. The timer was restarted once the recovery period had elapsed, regardless of whether the firefighter was in the starting position.	Full personal protective equipment and breathing apparatus.	Univariable and multivariable linear regressions were performed to determine the independent variables associated with test performance as an outcome. Standardized beta coefficients were used to interpret the strength of the association. Univariable and multivariable logistic regressions were performed to determine the independent variables associated with PAT pass rates. Backward stepwise linear regression models were performed to determine the factors contributing most to performance test completion time.

Legend: * AF = aerobic fitness; UBe = upper-body endurance; UBs = upper-body strength; LBe = lower-body endurance; LBs = lower-body strength; ANc = anaerobic capacity; ANp = anaerobic power; ABe = abdominal endurance; FL = flexibility; BF% = body fat percentage; mCAFT = modified Canadian aerobic fitness test; N/A = not applicable; NIOSH = National Institute for Occupational Safety and Health; RM = repetition maximum.

## Data Availability

Data that is not in the Appendix A will be shared if requested.

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
