# Peer review of "Association of Physical and Emotional Parameters with Performance of Firefighters: A Systematic Review"

_ijerph, 2024, doi:10.3390/ijerph21081097_

Round 1
Reviewer 1 Report
Comments and Suggestions for Authors
This paper provides a novel review of the association of physical and emotional parameters with performance of firefighters. The paper on the whole is well written with just a few suggestions of amendments from myself.
Introduction
Lines 73-76
I would recommend providing more detail for the reader on the definition of a tactical athlete and how the occupation of a firefighter could be categorised in this way. From the explanation you have given, readers could mistake a tactical athlete as an occupation with only high levels of stress and mental fatigue. This could include many occupations, some of which may not be physical (finance, law etc.). You define a tactical athlete briefly in the discussion section but it would be beneficial to put more detail in this section.
Methods
Line 96
What was the consensus agreement percentage set at? Do you have data on the agreements and disagreements between the reviewers.
Line 144
Who conducted the appraisal of the studies? If only one person why? And if two individuals, how were any disagreements dealt with?
Results
Line 212
Be more concise with your statements during the results section. Instead of saying “When it comes to lower body endurance…..”, I would suggest changing to “Two papers applied tests assessing lower body endurance…….”. At the moment, the current wording reads too conversational.
Discussion
Lines 260-262
You state that emotional and mental aspects are being heavily investigated in sports contexts and provide five references. What are the results from these studies? Is there a correlation (positive or negative) to support your proposed argument that mental health issues can decrease a firefighter’s performance? If the results from these studies show no correlation, then your next statement in line 262 “This comparison is pertinent” may be hyperbole if we, the readers don’t know what the results are from the studies you cited. This paper would benefit from providing correlations and P values (for example) from the papers cited.
Lines 287-290
This is much better. Providing correlation date and r values.
Line 311
Remove “the” from “All the eight studies…”
Line 337-339
Is there a P or R value for the significant correlation in the studies referenced.
Line 361-362
Need to be more concise. Provide the range of the r value not “around, this is too vague.
Line 381
Importance of power in what? Need to provide clear detail of what you a stating to ensure the reader understands what you mean.
Line 400
“…For less effort and better performance”. I would amend to physical effort for clarification. Especially as this paper discusses other efforts including mental health.
Author Response
Comment 1:
Lines 73-76
I would recommend providing more detail for the reader on the definition of a tactical athlete and how the occupation of a firefighter could be categorized in this way. From the explanation you have given, readers could mistake a tactical athlete as an occupation with only high levels of stress and mental fatigue. This could include many occupations, some of which may not be physical (finance, law etc.). You define a tactical athlete briefly in the discussion section but it would be beneficial to put more detail in this section.
Response 1: Thank you for your review. I agree and have included an entire paragraph on line 75 explaining the context.
Comment 2:
Line 96
What was the consensus agreement percentage set at? Do you have data on the agreements and disagreements between the reviewers.
Response: We consulted the records made by the two researchers and added in the methods part, thanks for the suggestion. Here we emphasize the text: “fourteen of the fifteen included studies were consensus between the two researchers, making an agreement of 93%. A total of eight papers, of the 142 screened, were selected as doubt by the two reviewers (six by VMF and two by EB). From these eight, after consulting the third reviewer, one of them was included.”
Line 144
Who conducted the appraisal of the studies? If only one person, why? And if two individuals, how were any disagreements dealt with?
Response: we adjusted the paragraph and added the requested data as follows: The examination was also conducted separately by two reviewers (VMF) and (EB). After which one rated the studies, a comparison was made and then discussed any discrepancy to reach an agreement. The initial ratings scored for each reviewer showed a consensus of 80%, being 12 papers equally classified.
Comment 3:
Line 212
Be more concise with your statements during the results section. Instead of saying “When it comes to lower body endurance…..”, I would suggest changing to “Two papers applied tests assessing lower body endurance…….”. At the moment, the current wording reads too conversational.
Response 3: We changed the sentence as suggested.
Comment 4:
Lines 260-262
You state that emotional and mental aspects are being heavily investigated in sports contexts and provide five references. What are the results from these studies? Is there a correlation (positive or negative) to support your proposed argument that mental health issues can decrease a firefighter’s performance? If the results from these studies show no correlation, then your next statement in line 262 “This comparison is pertinent” may be hyperbole if we, the readers don’t know what the results are from the studies you cited. This paper would benefit from providing correlations and P values (for example) from the papers cited.
Response 4: Thank you for your suggestion. We included examples, results, and P-values of the studies to better demonstrate the interaction between psychological variables and physical performance. We added the main results of the studies in the fourth, sixth, and seventh paragraphs, making this link clearer for the reader.
Comment 5:
Line 311
Remove “the” from “All the eight studies…”
Response 5: removed.
Comment 6:
Line 337-339
Is there a P or R value for the significant correlation in the studies referenced.
Response 6: We included the results of studies (R value) in the section and added the results in the next paragraph as well.
Comment 7:
Line 361-362
Need to be more concise. Provide the range of the r value not “around, this is too vague.
Response 7: altered as suggested.
Comment 8:
Line 381
Importance of power in what? Need to provide clear detail of what you a stating to ensure the reader understands what you mean.
Response 8: an explanation was added in order to provide better understanding.
Comment 9:
Line 400
“…For less effort and better performance”. I would amend to physical effort for clarification. Especially as this paper discusses other efforts including mental health
Response 9: altered as suggested.
Reviewer 2 Report
Comments and Suggestions for Authors
What is the rationale of this study? Why a systematic review is required in this topic. Have any previous experimental studies been conducted on these topics? If yes, this has to be reported in the literature review.
Registration details of the review should be separated from the search procedure.
The search is not based on PICOS criteria.
The search criteria of each database should be presented in separate tables or a supplementary file.
Why is the Pedro scale used for risk bias assessment
Certainty assessment is not performed
Risk factor assessment results should be presented separately in the results with tables.
The main characteristics table should be available in the main text
There is no proper discussion of results especially regarding the mental and emotional aspects in the discussion section.The authors only replicated the results in words
Add strengths, limitations and practical application of the study in the discussion section
Author Response
Comment 1:
What is the rationale of this study? Why a systematic review is required in this topic. Have any previous experimental studies been conducted on these topics? If yes, this has to be reported in the literature review.
Response 1: We add information as suggested
Introduction: we reinforce the purpose of the review by adding evidence of the correlational studies in the topic and explained better the relationship between psychological variables and its possible influence on FPATs performance. Besides, we highlighted the statement made by other authors that this theme has a poor number of systematic reviews. We also observed the different values of correlations encountered in previous literature and the discrepancy between the variables assessed. We believe that this systematic review can contribute to elucidate this theme. We hope that made ourselves more clear and that we justified the purpose for this study.
Comment 2:
Registration details of the review should be separated from the search procedure.
Response 2: We reallocated the registration details in the last sentence of the introduction part.
Comment 3:
The search is not based on PICOS criteria.
Response 3: we conducted the screening and selection of the papers based on PICO criteria as showed in section 2.2. Here we highlight the method:
- Population: Firefighters on active duty, whether volunteers or careers, were included. Recruits with no experience and civilians were excluded due to the bias of experience in job tasks.
- Intervention: articles that applied tests that measured performance in job-related firefighting tasks were included, with the parameters being time or other physiological factors. Excluded researches were evaluations in which firefighters didn't use complete PPE; used simulated weight vests, considering the bias of specific properties of the gear; applied tests only for one particular group of professionals such as wildland or pre-hospital care; used only one task, like only stair climb or only forcible entry, for example; applied live burn scenarios, since hot environments can influence.
- Comparison: included physical field or laboratory tests, physiological and anthropometric parameters, and emotional, mental/cognitive, or psychological variables in any measure.
- Outcomes: search included reports that showed values of statistical correlations, in any method, between the job performance tests and the comparison parameters.
Comment 4:
The search criteria of each database should be presented in separate tables or a supplementary file.
Response 4: A document with the search criteria has been added to supplementary material 1 (S1)
Comment 5:
Why is the Pedro scale used for risk bias assessment
Response 5: we included a justification in the text on the section 2.4: “Given this sclae has been widely used in previous reviews [20, 39, 40] as a validated tool, and was specifically designed for physiotherapy and physical performance studies, we judged it as particularly well-suited for evaluating research on the theme of this study, allowing a consistent quality assessment [38].”
Comment 6:
Certainty assessment is not performed
Response 6: in section 2.4 we included the certainty assessment, right after the justification for utilizing PEDro scale.
Comment 7:
Risk factor assessment results should be presented separately in the results with tables.
Response 7: A table of articles and values from the PEDro scale has been added to supplementary material 2 (S2)
Comment 8:
The main characteristics table should be available in the main text
Response 8: The table with the characteristics has been added to the main text
Comment 9:
There is no proper discussion of results especially regarding the mental and emotional aspects in the discussion section. The authors only replicated the results in words
Response 9: The discussion was broadened
Comment 10:
Add strengths, limitations and practical application of the study in the discussion section
Response 10: added as suggested.
Reviewer 3 Report
Comments and Suggestions for Authors
Dear authors,
The article titled "Association of physical and emotional parameters with performance of firefighters: a systematic review." demonstrates several strengths. It follows PRISMA guidelines and uses the PEDro scale for quality assessment. Furthermore, it provides recommendations for firefighter fitness programs and identifies current literature gaps. Despite these strengths, the paper has several weaknesses, including:
1. The study couldn't find significant correlations between psychological factors and FPAT performance.
2. The included studies have different testing protocols, which may affect the comparability of the study results.
3. Although live-burn scenarios are highly relevant to firefighters' performance, studies involving them were excluded from this study.
4. Unfortunately, the study does not present any new insights that do not exist in the reviewed literature.
Best regards,
A reviewer
Author Response
Comment 1:
- The study couldn't find significant correlations between psychological factors and FPAT performance.
Response 1: In fact, none were found because there were no studies that directly and acutely evaluated the influence of psychological and emotional variables on performance with the same procedures stated in this study. In our review, we found other papers that highlighted the theme, but with different methods of evaluating either performance and/or effects of psychological variables, then, it was inadequate with the proposed approach of this research. Most studies evaluated the influence of the day-to-day job of the firefighter in the psychological variables, and we were looking for the opposite relation. We added some of these studies in the discussion chapter to better elucidate the subject.
Comment 2:
- The included studies have different testing protocols, which may affect the comparability of the study results.
Response 2: We made every effort to standardize the tests to be as similar as possible, including tasks common to all firefighters, without specific tests for specialized groups. Worldwide, there are not many standardized tests for firefighters. For those that do exist, however, there are not many correlation studies that can form a solid basis for a robust analysis of the results. If the tests were strictly standardized, very few studies would be found. Given this situation, it was necessary to find similar tests to make an analysis possible. Despite different protocols, we aimed to use evaluations with the same foundation. We included the explanation in the limitations of the study.
Comment 3:
- Although live-burn scenarios are highly relevant to firefighters' performance, studies involving them were excluded from this study.
Response 3: Real fire scenarios were not used in order to standardize the tests as much as possible. If we had included tests with real fire, the factors of heat, sweat and dehydration would have significantly complicated the comparison between the circuits that did not use real fire. Besides, the fire per se is a difficulty factor for controlling and might vary a lot. Stating circuits that did not use fire was a manner of reducing bias in comparison. Indeed, this is a limitation of this study, but we judged it better to not mix the two kinds of tests. We added it in limitations section and explained.
Comment 4:
- Unfortunately, the study does not present any new insights that do not exist in the reviewed literature.
Response 4: For circuit correlations with physical tests, no major new findings were discovered. However, some important issues can be evaluated, as further elucidated below. Certain variables, such as short-duration anaerobic exercises and lower limb strength and endurance, seem not to have been fully studied or explored. These factors did not provide a sufficient body of evidence for conclusions and still offer avenues for further research regarding FPATs. As for psychological variables and their relationships with performance, very few studies were found that met the procedures of this research. This fact itself indicates a gap in the literature that could be better explored, given the influence these variables have shown on athletes' performance in the sports context, making this finding very interesting. We have added a more detailed explanation of the psychological issues in the conclusion section. Thank you very much for your suggestions; we hope that you have a better understanding of our intentions and that the paper has been improved based on the comments you provided.
Round 2
Reviewer 2 Report
Comments and Suggestions for Authors
The paper has been improved. However, it shows a high percentage of plagiarism
Author Response
Hello,
We are very happy that you are concerned about the quality of our document.
We ran a plagiarism analysis and no significant plagiarism was found in our document, something around 1%, but in the analysis of the document these points of plagiarism were not real, just quotes or terms. Of course, I understand the limitations of some detectors, and ours may have failed in this analysis, so if there is a document with these notes we will be happy to make the necessary adjustments.
I'll also send you the document with the plagiarism report
Thank you very much
